# Electrochemical DNA Sensors with Layered Polyaniline—DNA Coating for Detection of Specific DNA Interactions

**DOI:** 10.3390/s19030469

**Published:** 2019-01-24

**Authors:** Tatjana Kulikova, Anna Porfireva, Gennady Evtugyn, Tibor Hianik

**Affiliations:** 1A.M. Butlerov’ Chemistry Institute, Kazan Federal University, 18 Kremlevskaya Street, Kazan 420008, Russia; wefy2009@yandex.ru (T.K.); porfireva-a@inbox.ru (A.P.); 2Department of Nuclear Physics and Biophysics, Comenius University, Mlynska dolina F1, 842 48 Bratislava, Slovakia

**Keywords:** electrochemical DNA sensor, polyaniline, electropolymerization, DNA intercalator, doxorubicin, electrochemical impedance spectroscopy

## Abstract

A DNA sensor has been proposed on the platform of glassy carbon electrode modified with native DNA implemented between two electropolymerized layers of polyaniline. The surface layer was assembled by consecutive stages of potentiodynamic electrolysis, DNA drop casting, and second electrolysis, which was required for capsulation of the DNA molecules and prevented their leaching into the solution. Surface layer assembling was controlled by cyclic voltammetry, electrochemical impedance spectroscopy, atomic force, and scanning electron microscopy. For doxorubicin measurement, the DNA sensor was first incubated in the Methylene blue solution that amplified signal due to DNA intercalation and competition with the doxorubicin molecules for the DNA binding sites. The charge transfer resistance of the inner layer interface decreased with the doxorubicin concentration in the range from 1.0 pM to 0.1 μM (LOD 0.6 pM). The DNA sensor was tested for the analysis of spiked artificial urine samples and showed satisfactory recovery in concentration range of 0.05–10 μM. The DNA sensor developed can find application in testing of antitumor drugs and some other DNA damaging factors.

## 1. Introduction

DNA sensors are widely used for the determination of specific biochemical interactions important in medical diagnostics, food quality assessment, and pharmaceutics [1]. Most of the DNA sensors described are intended for detection of certain oligonucleotides related to specific pathogens and complementary to DNA probes attached to the biosensor interface [2]. Nevertheless, there is a growing interest in extension of their application to antitumor drug screening [3], DNA damage, and thermal inactivation detection, etc. Cytostatic drugs prevent DNA transcription in cancer cells due to DNA intercalation [4]. In such in vivo reactions, geometry, specific volume, flexibility, and other properties of DNA molecules are changed. This affects their biochemical functions [5]. Intercalators can also promote oxidative DNA damage via oxidation of guanine residues and formation of 8-oxoguanine, an indicator of oxidative stress [6,7]. For this reason, DNA sensors based on native DNA molecules are considered to be promising for detection of drug residues in biological liquids and for searching more effective and less toxic cytostatic drugs.

Detection of anticancer drugs with electrochemical biosensors is mainly based on the monitoring of intrinsic redox activity of DNA molecules immobilized on the electrode interface [7,8,9]. Thus, idarubicin was detected in the micromolar range of its concentrations by monitoring guanine oxidation at 1.1 V with cyclic voltammetry (CV) and differential pulse voltammetry (DPV) [8]. DNA immobilized onto reduced graphene oxide decorated with Au nanoparticles has been employed for quantification of interaction with doxorubicin by DPV technique using oxidation peak current promoted by the drug binding (limit of detection (LOD) 8 μg/mL) [10]. Various DNA intercalators have been detected with carbon nanotube printed electrode in the presence of single- and double-stranded DNA molecules based on nucleotide oxidation currents [11]. Among others, doxorubicin was determined in micromolar range of its concentrations. In similar conditions, interaction of DNA with doxorubicin resulted in reduced cathodic current measured with square wave and adsorptive stripping voltammetry and referred to adenine and cytosine [12]. Variation of the DNA sequence confirmed preferable binding of doxorubicin to guanine and adenine residues [13]. In some cases, own redox activity of drugs interacting with DNA was monitored [14,15,16]. However, in most cases, oxidation of nucleic bases and cytostatic drugs requires high potential and interferes with antioxidants present in the sample tested or drug medications as stabilizers. Besides this, many other species can affect the signal due to adsorption on the electrode surface.

Another way to monitor specific DNA interaction assumes changes in the permeability of the surface layer caused by implementation of intercalator and variation in charge separation of double stranded DNA molecules within the layer. Such a response can be recorded with electrochemical impedance spectroscopy (EIS) [17,18]. Contrary to the measurement of DNA-drug electroactivity, EIS makes it possible to quantify DNA-drug interaction for electrochemically inactive drugs. Rather low changes in the surface layer structure resulted from the big difference in the size of analyte and biochemical receptor molecules can be compensated by simultaneous changes in the hydrophobicity of the electrode interface or by the use of nanomaterials (Au nanoparticles, carbon nanotubes) that increase both specific surface concentration of biochemical receptor and layer density achieved after its saturation with the drug molecules.

Nevertheless, EIS measurements need to be improved in order to increase both sensitivity and selectivity of the signal toward target analytes. This is of special importance for the measurements performed directly in biological fluids, where proteins and excessive amounts of electrolytes can affect sorption and hence transfer of ferricyanide ions utilized as redox probe.

Inclusion of DNA molecules in polymeric films obtained by electropolymerization performed in the presence of biopolymer or prior to its adsorption can also significantly extend the capabilities of detection of DNA-drug interactions [19]. Polyaniline (PANI) and its derivatives are most frequently used for this purpose. They are synthesized in the presence of strong acids by addition of some oxidants ((NH_4_)_2_S_2_O_8_, FeCl_3_, or K_2_Cr_2_O_7_) [20,21,22,23,24]. Addition of polyelectrolytes and anionic surfactants accelerates the formation of oligomeric products due to stabilization of positively charged intermediates [25]. In electrochemical sensors, PANI is mostly synthesized by multiple cycling of the electrode potential in the monomer solution [26]. The reaction requires strong acidic conditions [27]. The properties of the polymer obtained depend on the content of working solution, potential range, counter anion, and electrode pre-treatment. The interest to PANI covered electrodes is mostly caused by its redox and electrocatalytic activity and electroconductive properties exerted mostly at pH < 3 [28]. In biosensor assembly, PANI can be used for generation of the signal to the H^+^ ions released in enzymatic reactions [29,30] or as support for immobilization of proteins or DNA probes [31,32]. DNA as polyanion stabilizes positively charged semi-oxidized emeraldine form of PANI and hence can be used as template for PANI formation [33,34,35]. Nevertheless, templating effect does not compensate fully for DNA damage caused by strong acids required for polymerization.

The use of organic acids instead of sulfuric and hydrochloric acids commonly used for aniline polymerization allows decreasing concentration of H^+^ ions on the stage of electrosynthesis without negative consequences for polymer redox activity [36]. Recently we have described DNA sensors with electrochemical polymerization of aniline performed in oxalic acid in the presence of prior to sorptional application of native DNA. It was shown that reduced acidity of reaction media preserved ability of native DNA to interact with intercalators including anthracycline antitumor drugs. In this work, we suggest to prepare layered coatings of biosensor with DNA entrapped between two independent PANI layers obtained by electropolymerization. Deposition of the second PANI film suppressed leaching DNA from the surface and preserved its configuration most comfortable for target interactions.

## 2. Materials and Methods

Doxorubicin, Methylene blue (1), and low-molecular DNA from salmon sperm (Cat. No. 31149, average mol. mass 4.6 kDa [37]) were purchased from Sigma-Aldrich (Sigma-Aldrich, St. Louis, MO, USA). Aniline (Alfa Aesar, 99+%, Karlsruhe, Germany) was distilled under vacuum and stored under argon prior to use. All the other reagents were of analytical grade. All the measurements were performed in Britton-Robinson buffer prepared from 0.04 M acetate acid, 0.04 M phosphoric acid, and 0.04 M boric acid with 0.05 M sodium sulfate adjusted to pH 3.0). Artificial urine samples contained 10 mM CaCl_2_, 6 mM MgCl_2_, 80 mM NaCl, 16 mM Na_2_SO_4_, 2 mM potassium citrate, 20 mM KH_2_PO_4_, 21 mM KCl, 18 mM NH_4_Cl, 9 mM creatinine, and 416 mM urea. All the reagents were of analytical grade. Deionized Millipore^®^ water (Millipore Simplicity^®^, Merck, Darmstadt, Germany) was used for preparation of working solutions.



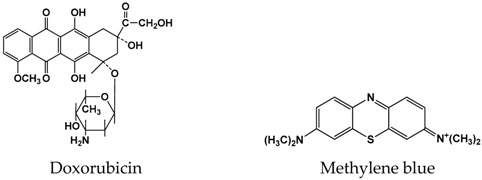
(1)


Voltammetric measurements were performed using the CHI Electrochemical Workstation 660E (CH Instruments Inc., Austin, TX, USA). Home-made glassy carbon electrode (GCE) consisted of a 2 cm rod with the geometric surface area of the cut end equal to 1.67 mm^2^. The rod was pressed in the polytetrafluoroethylene tube and mechanically polished to mirror-like surface. On the opposite side, stainless steel thread connection was attached as a current collector. Pt wire was used as auxiliary electrode and Ag/AgCl (1.0 M KCl) as reference electrode. The EIS spectra were recorded at the formal potential of ferricyanide redox probe defined as half-sum of its cathodic and anodic peak potentials. The amplitude of the applied sine potential was equal to 5 mV and frequency varied from 100 kHz to 0.04 Hz. Fitting procedure and calculation of the capacitance and the charge transfer resistance were performed from the Nyquist diagrams using NOVA software (Metrohm Autolab). *R*(*C*_1_*R*_1_)(*C*_2_*R*_2_) equivalent circuit was used for data fitting where *R* is resistance and *C* is CPE, which is equal to the capacity. The indices 1 and 2 correspond to the inner and outer interface of the electrode-modifier layer system.

The atomic force microscopy (AFM) images were obtained with the Bruker FastScan Dimension Bio (Bruker, Karlsruhe, Germany) in tapping mode with a 90 × 90 μm scanner and silicon cantilevers. The scanning electron microscopy (SEM) images were obtained with the high-resolution field emission scanning electron microscope Merlin™ (Carl Zeiss, Oberkochen, Germany).

Electropolymerization of aniline was performed by continuous cycling of the GCE potential between −0.3 and 1.3 V at 50 mV/s in solution containing 0.07 M aniline in 0.5 M oxalic acid. After that, the electrode was washed twice with deionized water and fixed upside down. An amount of 20 μL of 1 mg/mL DNA solution were placed on its working surface. The electrode was capped with plastic tube to prevent drying and left for 15 min. Then it was gently washed again and put in the same working solution to repeat electropolymerization step. In most experiments, 4 cycles of potentials were applied both prior to and after DNA application. After surface layer assembling, the electrode with layered PANI-DNA-PANI coating was incubated in MB or doxorubicin solution for 10 min, washed and moved in Britton-Robinson buffer, pH 3.0, for recording cycling voltammograms or for EIS measurements.

## 3. Results and Discussion

### 3.1. Surface Layer Assembling

Figure 1a shows the first 4 cycles on voltammogram recorded in 0.5 M oxalic acid containing 0.07 M aniline. As could be seen, the polymerization reaction is initiated by formation of irreversible anodic peak at a high anodic potential (about 0.9 V). It sharply decayed with number of cycles indicating coverage of the surface with polymer. Besides irreversible peak, cyclic voltammogram contained a broad reversible peak pair in the range from 0.2 to 0.6 V commonly attributed to the internal redox conversion of PANI forms. Similar shape of the peaks was also observed in 0.5 M sulfuric acid whereas 1 M HCl and HClO_4_ demonstrated a number of peak pairs related to formation and oxidation of semi-oxidized emeraldine salt exerting electroconductive properties of PANI [28]. Thus, the formation of a single broad peak can be explained by moderate acidity of oxalic acid used in this work.

A broad shape of reversible peak pair masked its increase with the number of potential runs typical for aniline electropolymerization. Current changes are mostly located at 0.1–0.3 V, i.e., before fast increase of the current near the bottom of main peaks. This might be a result of accumulation of quinoidal by-products formed at high anodic potential. They could partially suppress the following polymerization due to involvement in reversible electron exchange. As was established earlier for the PANI obtained in sulfuric acid, decrease of the upper potential at the multiple potential scanning to 0.8–0.9 V minimized the formation of such by-products. However, experiments performed earlier [36] indicated insufficient accumulation of PANI formed in the presence of oxalic acid in such conditions. 

It is interesting to compare the voltammograms recorded in the same conditions before and after intermediate adsorption of native DNA onto freshly deposited PANI layer. The DNA deposition was performed after removal of excessive acid by washing with deionized water to prevent acidic damage of the biopolymer. The comparison of voltammograms (Figure 1a,b) confirms inclusion of DNA in the surface layer. The PANI peaks became higher and narrower than before DNA adsorption. The peak pair at 0.2–0.3 V became more pronounced and increased faster than prior to DNA addition. Below, the electrode covered with such coating is denoted as GCE/PANI/DNA/PANI. Similar measurements performed at higher pH value indicate a sharp decay of the peaks attributed to PANI emeraldine form [28] due to its conversion into leuco-emeraldine with not electroconductivity and redox-activity required for signal transduction. However, at lower pH (below pH 5) the protonation of DNA can change their properties. Certainly, it has been shown that pH changes from 4 to 3 resulted in unstacking of N-7 guanine which rotating out of the helix [38]. Nevertheless, investigation of the effect of protonation of DNA (at pH between 4 and 3) on the interaction of Nitidine chloride clearly demonstrated intercalation of this compound inside the helix [39]. It is also possible that that the entrapment of DNA between aniline layers stabilize the DNA double helix.

When moved from polymerization solution to the weakly acidic solution (pH 3.0) with no aniline, single symmetric peak pair was observed. Their height and position depended on the presence of low-molecular compounds able to interact with DNA (Figure 2). They were tested by incubation of the DNA sensor followed by its transfer to the fresh working buffer and recording cyclic voltammogram in the absence of analytes in the solution. 

Thus, incubation of the biosensor in the solution of Methylene blue, a redox active intercalator, improved the conditions for electron exchange on the electrode interface. Its incorporation in intermediate DNA layer makes the peaks more symmetrical due to significant increase of the cathodic current on the reversed branch of voltammogram (curve 2). Meanwhile, no own peaks of the dye were found because free molecules of Methylene blue were forced away the coating due to electrostatic repulsion of positively charged PANI and drug molecule. If the electrode was consecutively incubated in doxorubicin solution, Methylene blue molecules were competitively released from the DNA helix and participated in the redox reaction of inner PANI layer. As a result, the appropriate peak currents increased (Figure 2, line 3). Thus, DNA capped with PANI thin film retains its ability to intercalate with anthracycline and phenothiazine drugs. The formation of PANI inner layer was also confirmed by AFM (Figure 3). 

During the first 4 cycles of the potential, the polymer formed rounded particles distributed among the surface preferably in the GCE defect areas and scratches (Figure 3a). 

The formation of PANI layer only partially covering GCE surface was confirmed for first step (4 cycles) by cyclic voltammetry of ferricyanide redox probe using its cathodic peak current. To exclude mediation of the electron transfer aided by PANI, ferricyanide cyclic voltammogram was recorded at pH 7.0 where PANI is inactive. Ratio of free accessible and geometric area of the electrode was calculated from the peak current using the Randles-Sevcik equation and was found to be 0.75.

After 8 cycles of potential scanning, more uniform dense film was obtained with few particles randomly covering the surface with no orientation along the scratches. Maximal difference of the coating thickness decreased with higher number of potential cycles about twice. 

The adsorption of DNA onto PANI layer and its coverage the polymer in following potential cycling were confirmed by SEM (Figure 4). DNA forms elongated aggregates consisted of particles different in size and shape that are distributed onto the even support surface. Deposition of PANI onto the DNA resulted in formation of a rough surface with particles both corresponded to the shape of DNA aggregates and those much lower in size. This means DNA remains adsorbed on the surface and stimulate electropolymerization of new PANI molecules by template effect.

Deposition of PANI onto DNA is explained by template effect described in the literature [40,41,42]. Positively charged oligomers and protonated aniline molecules are accumulated onto phosphate residues of the DNA backbone. This results in formation of elongated PANI particles (wires or pins) that are utilized for electric wiring of oxidoreductases or connections in semiconductor devices. DNA template effect also increases average roughness of the support and specific surface of the modified electrode.

### 3.2. Voltammetric Determination of Doxorubicin

As could be seen from preliminary experiments, incubation of the DNA sensor in doxorubicin solution changes the ratio and value of anodic and cathodic peak currents attributed to the internal PANI activity. This might be a result of DNA intercalation followed by changes in the flexibility of the DNA molecules, its volume and distance between negative phosphate residues. Meanwhile, this influenced the PANI activity rather modestly probably due to lack of doxorubicin redox activity in the range of potentials measured. The consecutive treatment of the DNA layer with Methylene blue and doxorubicin increased changes observed due to competition between both molecules for the same binding sites of DNA. 

Methylene blue intercalates DNA helix in the area reach with guanine residues. Besides this, it can be electrostatically accumulated near minor grooves of DNA sequence by interaction with phosphate groups. Such interactions differ its electrochemical activity that can be monitored using conventional electrochemical tools. Mechanism and examples of application of Methylene blue in DNA sensors are considered in [43,44,45].

Methylene blue released from the DNA helix is involved in the chain of the electron transfer and hence increases the currents recorded. It should be noted that the position and shape of the PANI peak do not allow separating contribution of direct oxidation of Methylene blue of the electrode. However, taking into account ultra-low concentrations of the dye that could be accumulated in such experiment, this alternative mechanism should be regarded as much less probable. Hence, the following determination of Methylene blue was performed in the following conditions: the DNA sensors were first incubated in 0.5 mM Methylene blue solution for 15 min, then washed in working buffer, pH 3.0, incubated in doxorubicin solution for the same time and moved again in fresh buffer solution to record cyclic voltammogram. For doxorubicin quantification, changes in cathodic peak current at about 0.05 V were used because of higher changes and better reproducibility against anodic peak current.

The conditions of doxorubicin determination were optimized to reach maximal sensitivity and lowest LOD (Table 1). The LOD value was calculated for S/N = 3 criterion. The number of potential cycles applied prior to DNA application was chosen to be the same (4 cycles). It was determined from the necessity to reach maximal roughness of the PANI layer and to leave a part of bare electrode for further DNA adsorption. The DNA sensors obtained with 6 cycles of potential cycling performed in the presence of DNA have been recently described either [46]. First, the amounts of DNA deposited in between the PANI layers were varied. Although DNA adsorption was governed by electrostatic interactions and could depend mostly on the underlying layer properties, the increase of nominal DNA concentration applied affected the sensitivity of doxorubicin determination.

Higher amounts of DNA applied increased the slope of calibration curve probably due to template effect of DNA stimulating deposition of the second PANI layer and hence sensitivity of alteration of its redox properties. Besides this, higher quantities of biopolymer accumulate more intercalators and increase the deviation of the signals recorded. Similar influence was found for increasing number of layers of PANI deposited onto the DNA. Meanwhile the LOD estimated from S/N = 3 criterion was about the same for 4, 6, and 10 cycles of PANI deposition on the DNA layer (see Table 1). The following experiments were performed with deposition of 20 μL, 2.0 mg/mL DNA covered with PANI synthesized in 4 potential cycles.

The incubation period of separate steps of signal measurement altered the results of doxorubicin determination to a much lower extent. Thus, increase of the incubation of the DNA sensor in Methylene blue from 15 to 45 min. changed absolute values of the peak currents and the shift of the signal after the contact with 10 nM doxorubicin by 3–5%. This is comparable with standard deviation of the signal measurement. Variation of the period of the contact with doxorubicin from 10 to 60 min decreased the cathodic peak current by 3%. The extension of the incubation to 90 min paradoxically decreased both absolute value of the signal toward doxorubicin and the slope of appropriate calibration curve due to partial leaching of the Methylene blue from the surface layer. 

No specific requirements have been found for the use of Methylene blue solution, which can be prepared in advance and stored at room temperature for more than 2 weeks. To avoid mistakes, freshly prepared Methylene blue stock solution should be filtered to remove solid particles and impurities that could alter the dye concentration in contact with the biosensor. 

Specificity of the response was assessed by changing doxorubicin with sulfamethoxazole and native biopolymer with thermally denatured DNA. In case of sulfamethoxazole able to electrostatically interact with DNA helix onto minor grooves, the appropriate peak currents decreased as in the case of doxorubicin. This might be due to changes in electrostatic interactions affecting Methylene blue participation in the electron transfer chain on the electrode interface. Meanwhile, sulfonamides are unable to intercalate DNA and their influence on appropriate reactions of phenothiazine dye was quite moderate. As a result, sensitivity of the signal shift toward sulfamethoxazole was threefold lower than that toward doxorubicin taken in the same concentration range (0.22–0.25 μA/pC).

### 3.3. EIS Determination of Doxorubicin

Voltammetric measurements made it possible to conclude that changes in the peak currents attributed to the PANI layer can be due to influence of both DNA and its intercalators on the efficiency of electron transfer within the layer. In this case, EIS offers additional opportunities to quantify these effects and increase sensitivity of doxorubicin determination. Experiments were performed in the presence of [Fe(CN)_6_]^3−/4−^ redox probe at the formal redox potential of ferricyanide ion determined as half-sum of its peak potentials recorded on modified GCE. Appropriate Nyquist diagrams contain semicircle fragments that represent limitation of the charge transfer on inner and outer interfaces of surface layer (Figure 5). The equivalent circuit used for EIS data fitting is presented in (2). Here, *R*_s_ and *R_et_* are diffusion and charge transfer resistances, respectively, and *C* is the constant phase element. The roughness parameter *n* was found to be 0.88–0.92 indicating that CPE was equal to capacitance of appropriate interfaces. Index ‘1’ corresponds to the ‘electrode-film layer’ interface whereas index 2 to the ‘modifier-solution’ interface.
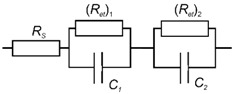
(2)

Prior to contact with doxorubicin, the charge transfer resistance irregularly responded to the surface layer assembling (Figure 6). Thus, deposition of the first PANI layer increased the *R_et_* value of the electrode-modifier interface from 1.2 to 1.6 kΩ. The following casting of DNA (20 μL, 2 mg/mL solution) decreased this value to 1.3 kΩ due to stabilization of emeraldine salt by negatively charged DNA molecules. Deposition of the second PANI layer onto adsorbed DNA increased the quantities of the PANI and capsulated adsorbed DNA molecules. This improved the conditions of the electron transduction and hence decreased the *R_et_* value to 0.65 kΩ. The following treatment with growing concentrations of doxorubicin gradually increase the resistance due to substitution of Methylene blue in the layer. The appropriate calibration curve is linear within the concentration range from 1.0 pM to 0.1 μM (Equation (3)). Mean ± standard error values were calculated for the slope and intercept of the curve. All the experimental points have been obtained in triplicate using individual sensors and analyte solutions.
*R_et_*, kΩ = (−0.26 ± 0.06) + (0.32 ± 0.01) × p*C*, M, *R*^2^ = 0.9942, n = 7(3)

In comparison with voltammetric measurements (Table 1), the LOD of doxorubicin was reduced by more than two orders of magnitude (LOD 0.6 pM). It should be also noted that *R_et_* relative changes on the upper interface are significantly lower and to some extent irregular and do not coincide with the analyte concentration. This confirms the idea that the changes in the EIS spectra are related to the DNA molecules capsulated in the middle of the surface layer. Similar measurements with no Methylene blue treatment (Figure 6b) did not show regular changes of the parameters because doxorubicin being free or incorporated did not exert its ability to electrochemical wiring of the DNA molecules. In both cases, DNA sensors provides a single measurement because the equilibria with participation of capsuled DNA molecules are shifted toward intercalated molecules and their dissociation within a reasonable time interval seems impossible.

The sensitivity of the impedimetric DNA sensor developed was higher than that reported for other DNA sensors. Appropriate comparison is presented in Table 2. 

A significant improvement of the LOD value has been obtained due to dense contact of the DNA molecules with the PANI in the surface layer and its higher influence on intrinsic redox activity of the polymer matrix. Besides this, contrary to the electropolymerization of aniline in the presence of DNA molecules, layer-by-layer immobilization allows choosing the conditions for DNA application and avoiding its contact with strong acids necessary for PANI electrochemical activity. 

Besides high sensitivity, the DNA sensors with layered PANI-DNA coating show some other advantages, e.g., more reliable protocol of the DNA immobilization, easy control of the surface layer assembling, choice of favorable pH independently for aniline electropolymerization, DNA application, and DNA-analyte interaction. For recent review on DNA-drug interactions, see also the paper by Fojta et al. [50].

### 3.4. Measurement Precision and DNA Sensor Lifetime

Measurement precision was assessed using six DNA sensors prepared from the same set of reagents within a day and tested for charge transfer resistance and cathodic peak current. The measurement-to-measurement repeatability calculated from cyclic voltammetry was equal to 2.5% and that for EIS protocol 3.4% (six measurements). Sensor-to-sensor repeatability of the same series was equal to 3.2% and 4.5%. The repeatability of relative shift of the *R_et_* value calculated for 10 nM doxorubicin was found to be 6.6%. 

The DNA sensor stored in dry conditions at 4 °C retains its voltammetric response for at least two months during which the cathodic peak current decreased by 20% of initial value but the response toward 10 nM doxorubicin only by 10%. When used after storage, the DNA sensor should be first incubated in Britton-Robinson buffer, pH 3.0, for 30–60 min. The incubation period can vary depending on the air humidity and storage temperature. The recovery of the biosensor is monitored by peak currents recorded prior to the contact with the drug solution and peak shape.

### 3.5. Real Sample Analysis

Applicability of the DNA sensor was confirmed by experiments with the samples of artificial urine spiked with doxorubicin. Only pH correction was required for sample testing. As was shown, lower limit of quantification (S/N = 10) increased against standard aqueous drug solutions to 10 nM and the LOD to 2 nM. The recovery varied from 102 to 110% within concentration range between 0.05–10 μM. Increased deviation of the signal can be attributed to the buffering properties of the urine samples and non-specific adsorption of some components on the PANI layer followed by partial suppression of its redox activity. Regarding real urine samples, it is believed the performance of the DNA sensor will be similar assuming higher deviation of the results caused from the personal variability of the urine content. A higher number of samples and measurements should be made to take into account possible complications and ensure patients in reliability of the results.

## 4. Conclusions

The implementation of the DNA molecules between two electropolymerized PANI layers made it possible to detect specific interaction of the biopolymer with some drugs (doxorubicin, Methylene blue) due to its influence on redox equilibria of PANI environment. Changes in the charge separation and partial compensation of negative charge of phosphate residues of DNA helix affect the amounts of emeraldine salt responsible for electrochemical signal measured by cyclic voltammetry or EIS. This offers opportunities to determine electrochemically inactive compounds. Separation of the stages of aniline polymerization and DNA deposition simplify assembling of the surface layer and allow changing pH of separate steps to get optimal conditions and hence higher sensitivity of the response on the whole. For this reason, sensitivity of doxorubicin determination was found to be higher than in case of simultaneous deposition of PANI and DNA from the same solution performed in a single stage [36]. The use of mediation properties of Methylene blue together with its competition with analyte for DNA binding sites improved the performance of DNA sensor. Such an effect can be further used for detection of other factors affecting DNA configuration, e.g., oxidative damage or thermal denaturing. Applicability of the DNA sensor developed was confirmed by testing spiked samples of artificial urine showing acceptable recovery and accuracy of signal measurement in urine after its pH correction.

## Figures and Tables

**Figure 1 sensors-19-00469-f001:**
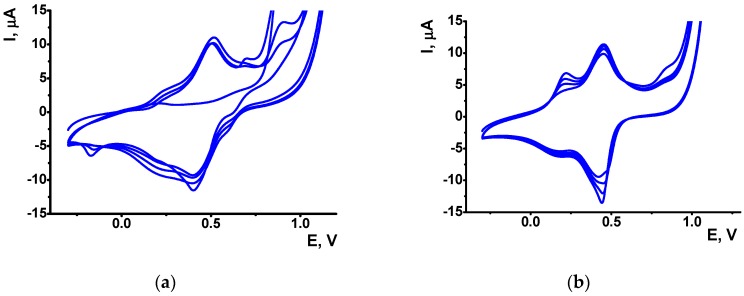
Aniline electropolymerization by multiple cycling of the GCE potential in 0.5 M oxalic acid containing 0.07 M aniline prior to (**a**) and after (**b**) the adsorption of DNA on the electrode surface. Scan rate 50 mV/s.

**Figure 2 sensors-19-00469-f002:**
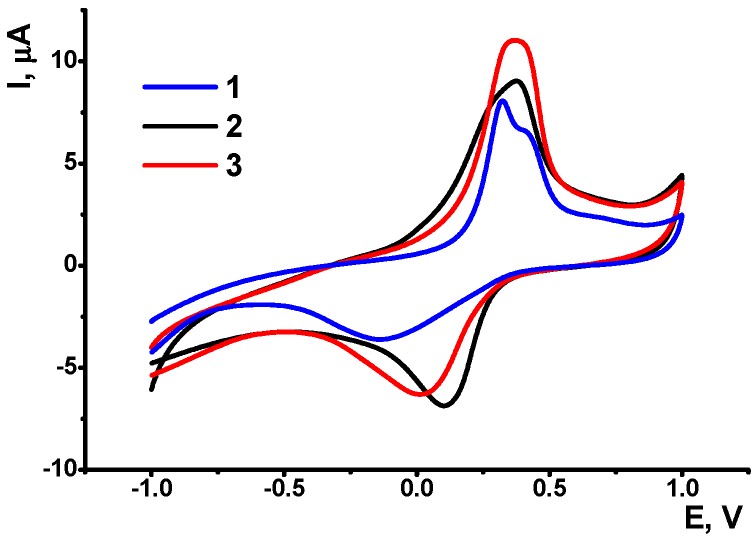
Cyclic voltammograms recorded on GCE/PANI/DNA/PANI prior to (1) and after consecutive 15 min incubation in 0.5 mM Methylene blue (2) and 10 nM doxorubicin (3).

**Figure 3 sensors-19-00469-f003:**
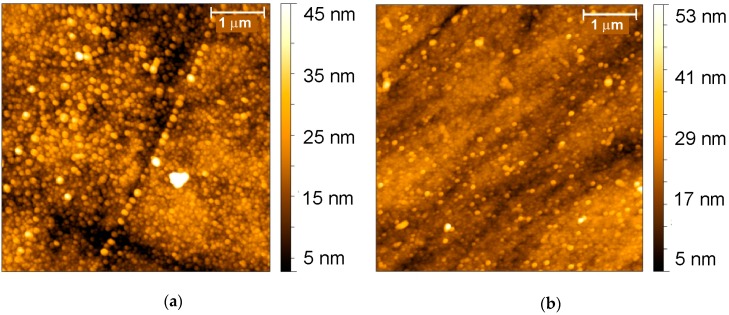
AFM images obtained on polished glassy carbon electrode (GCE) in four (**a**) and eight (**b**) cycles of potential in 0.5 M H_2_C_2_O_4_ containing 0.07 M aniline.

**Figure 4 sensors-19-00469-f004:**
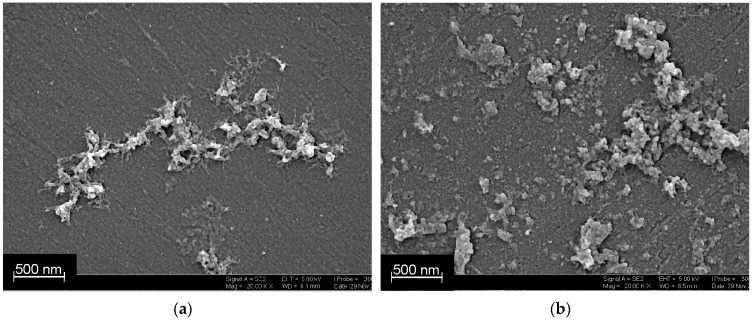
SEM images obtained on polished GCE covered with PANI (4 cycles) and adsorbed DNA (**a**) and PANI–DNA–PANI coating (**b**).

**Figure 5 sensors-19-00469-f005:**
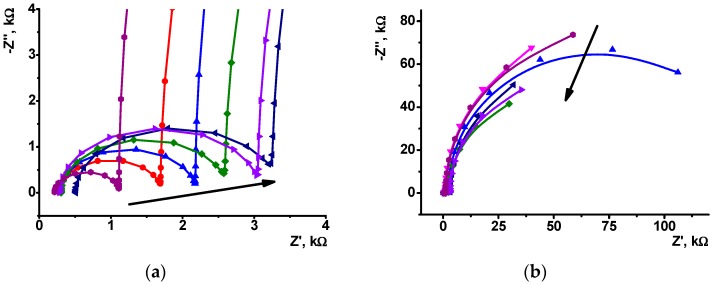
The Nyquist diagram of impedance spectra corresponded to inner (**a**) and outer (**b**) interface of the GCE/PANI/DNA (20 μL, 2 mg/mL)/PANI electrode consecutively incubated in 0.1 mM Methylene blue and 0, 1 × 10^−5^, 1 × 10^−6^, 1 × 10^−7^, 1 × 10^−8^, and 1 × 10^−9^ M doxorubicin. Measurements in the presence of 0.01 M K_3_[Fe(CN)_6_] and 0.01 M K_4_[Fe(CN)_6_] at 0.26 V vs. Ag/AgCl. Frequency range 0.04 Hz–100 kHz, amplitude 5 mV. Arrows indicate direction of consecutive increase of the doxorubicin concentration.

**Figure 6 sensors-19-00469-f006:**
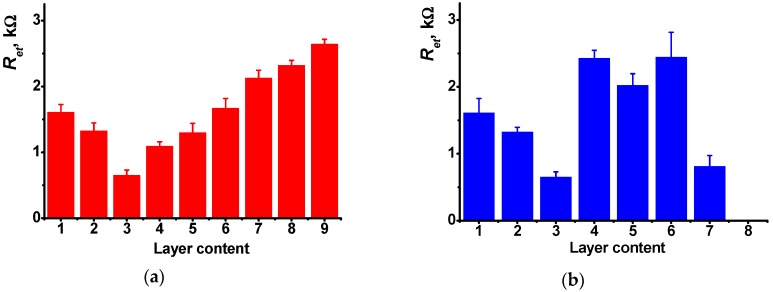
Charge transfer resistance of the inner ‘electrode-modifier’ interface of the GCE (1) covered with PANI (4 layers), (2) DNA (20 μL, 2 mg/mL), (3) PANI-DNA-PANI (4 layers), (4) 1 × 10^−12^ M, (5) 1 × 10^−11^ V, (6) 1 × 10^−10^, (7) 1 × 10^−9^, (8) 1 × 10^−8^, (9) 1 × 10^−7^ M doxorubicin. The results obtained with preliminary treatment of the electrode with Methylene blue (**a**) or without this step (**b**). All the measurements in 0.025 M universal buffer, pH 3.0. Average S.D. for three replications.

**Table 1 sensors-19-00469-t001:** Voltammetric determination of doxorubicin in different conditions of surface layer assembling, n = 7.

Parameter Changed	Parameter Value	Sensitivity of Doxorubicin Determination, μA/pC	Doxorubicin Concentration Range, M	LOD, nM
DNA application (4 PANI cycles)	20 μL, 0.5 mg/mL	0.45 ± 0.03	2 × 10^−8^–6 × 10^−3^	10
	20 μL, 1.0 mg/mL	0.58 ± 0.02	5 × 10^−8^–1 × 10^−3^	2.0
	20 μL, 2.0 mg/mL	0.79 ± 0.03	1 × 10^−8^–2 × 10^−4^	3.5
PANI cycles ^1^	2	0.55 ± 0.04	1 × 10^−8^–7 × 10^−3^	3.0
(20 μL of 1.0 mg/mL DNA)	6	0.61 ± 0.02	6 × 10^−9^–1 × 10^−4^	2.0
	10	0.70 ± 0.05	7 × 10^−9^–5 × 10^−5^	2.0

^1^ Number of potential cycles for aniline electropolymerization applied after the DNA adsorption.

**Table 2 sensors-19-00469-t002:** Comparison of analytical characteristics of doxorubicin determination with electrochemical sensors and DNA sensors.

Surface Modification	Signal Mode	Concentration Range	LOD, nM	Ref.
Fe2Ni@Au nanoparticles on reduced graphene oxide	Differential pulse voltammetry	0.003–5 mg/mL (5.5 µM–9.2 mM)	14,600	[10]
PANI/DNA	Cyclic voltammetry	0.1 nM–0.2 mM	0.01	[36]
Multiwalled carbon nanotubes	Cyclic voltammetry	0.09–7.36 µM	3	[47]
Multiwalled carbon nanotubes/polylysine	Square-wave voltammetry	2.5 nM–0.25 µM	1	[48]
Poly(Neutral red)/pillar [5] arene/DNA	Differential pulse voltammetry, EIS	0.01–100 µM	0.1	[49]
PANI/DNA/PANI	EIS	1.0 pM–0.1 mM	0.0006	This work

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
