# Peer review of "Electrochemical DNA Sensors with Layered Polyaniline—DNA Coating for Detection of Specific DNA Interactions"

_sensors, 2019, doi:10.3390/s19030469_

Round 1
Reviewer 1 Report
In this manuscript, Interaction of doxorubicin and methylene blue with DNA modified polyaniline surface has been studied with the support of several characterizations including CV, EIS, SEM, and AFM. The authors demonstrated the new approach that the DNA immobilized between two polaynailine films to avoid the loss of DNA molecule in the electrode surface. The present work is more appropriate for Sensors Journal. However, there are several clarification and modification which need to address for better clarity. Therefore, I recommend a minor revision to the submitted work.
The authors written as “electrochemical DNA sensor” in the manuscript title. Here, DNA is not target analyte, Just DNA modified surface served to study the detection of doxorubicin and methylene through interaction. So, the title should be revised.
2. What is the DNA sequences used in the work?
3. It is well-known that the DNA molecules well stable in physiological pH condition. But, in this work pH 3.0 was for the interaction study. What is the reason? Is it promote the DNA denature.
4. In the results and discussion part, the DNA modified surface has narrow and clear peaks during the polyaniline electro-deposition when compared with that of pre-polymerization on GC electrode. Here, the authors should explain the reason for peak enhancement and the role of DNA in polymerization process.
5. What are the major interaction played between DNA and MB. It should be explained with literature support ex. Anal. Methods 5 (2013) 1010-1015, Title “Polypyrrole nanotube-polyaniline composite for DNA detection using methylene blue as intercalator”.
6. In the experimental section, the authors mention that the aniline concentration as 0.07 M. However, in the results and discussion, the authors say 7 mM of aniline. What is correct?
The author’s need to provide detailed information about the importance of polyaniline and their properties for different applications in the introduction part with support of the following few references (Sensors 2018, 18(11), 3926 and Indian Journal of Chemistry 50A (2011) 970-978).
In addition, there are several mistakes and typical errors, and kindly double-check the manuscript for example few:
Original: Methylene blue ; Revised: methylene in Line number 19
Original: utilized ad Revised: utilized as in Line number 73
Author Response
We are grateful to the reviewer for most useful comments that allowed us to improve manuscript. All comments were taking into account in revised manuscript. The changes made are highlighted by red color. Response to the reviewer comments is below.
Comment 1:
The authors written as “electrochemical DNA sensor” in the manuscript title. Here, DNA is not target analyte, Just DNA modified surface served to study the detection of doxorubicin and methylene through interaction. So, the title should be revised.
Response:
Classification of biosensors is based on the nature of biocomponent applied in their assembly (enzyme sensor, immunosensor, aptasensor). In this sense, the title correctly describes the work, in which DNA is implemented in the surface layer and interacts with analytes intercalating DNA helix. Such understanding of DNA sensor term coincides with the recommendations of IUPAC (Technical report “Electrochemical nucleic acid-based biosensors: Concepts, terms, and methodology”, Pure Appl. Chem. 82 (2010) 1161-1187. We would therefore prefer to keep the title without changes.
Comment 2:
What is the DNA sequences used in the work?
Response:
Low molecular weight DNA from salmon sperm was used. Its mol. mass and reference to other characteristics were added to Experimental section as follows:
"… low-molecular DNA from salmon sperm (Cat. No. 31149, average mol. mass 4.6 kDa [37]" (Page 3, lines 101-102)
Comment 3:
It is well-known that the DNA molecules well stable in physiological pH condition. But, in this work pH 3.0 was for the interaction study. What is the reason? Is it promote the DNA denature.
Response:
The choice of pH was related to the properties of polyaniline, which is present in electroconductive form called emeraldine salt, only in acidic media. For this reason, higher pH results in worse conditions of electron exchange on the electrode interface. The reviewer is right concerning possible denaturation of DNA at lower pH (below pH 5). Certainly, it has been shown that pH changes from 4 to 3 resulted in unstacking of N-7 guanine which rotating out of the helix [38]. Nevertheless, detailed investigation of the effect of protonation of DNA (at pH between 4 and 3) on the interaction of Nitidine chloride clearly demonstrated intercalation of this compound inside the helix [39]. It is also possible that that the entrapment of DNA between aniline layers stabilize the DNA double helix. The following sentences were added top the Results and discussion section (Page 4, line 165).
"Similar measurements performed at higher pH value indicate a sharp decay of the peaks attributed to PANI emeraldine form [28] due to its conversion into leuco-emeraldine with not electroconductivity and redox-activity required for signal transduction. However, at lower pH (below pH 5) the protonation of DNA can change their properties. Certainly, it has been shown that pH changes from 4 to 3 resulted in unstacking of N-7 guanine which rotating out of the helix [38]. Nevertheless, detailed investigation of the effect of protonation of DNA (at pH between 4 and 3) on the interaction of Nitidine chloride clearly demonstrated intercalation of this compound inside the helix [39]. It is also possible that that the entrapment of DNA between aniline layers stabilize the DNA double helix."
Comment 4:
In the results and discussion part, the DNA modified surface has narrow and clear peaks during the polyaniline electro-deposition when compared with that of pre-polymerization on GC electrode. Here, the authors should explain the reason for peak enhancement and the role of DNA in polymerization process.
Response:
Increase of the PANI peaks is attributed to the template effect of DNA observed for both electrochemical and chemical PANI synthesis. Briefly, positively charged molecules of protonated aniline and its oligomers are electrostatically accumulated on negatively charged phosphate residues of the DNA chain and hence for particles with higher content of oxidized (positively charged) form of PANI and higher specific surface (higher roughness). The following text was added to the Section 3.1 (lines 201-206):
"Deposition of PANI onto DNA is explained by template effect described in the literature [40-42]. Positively charged oligomers and protonated aniline molecules are accumulated onto phosphate residues of the DNA backbone. This results in formation of elongated PANI particles (wires or pins) that are utilized for electric wiring of oxidoreductases or connections in semiconductor devices. DNA template effect also increases average roughness of the support and specific surface of the modified electrode."
Comment 5:
What are the major interaction played between DNA and MB. It should be explained with literature support ex. Anal. Methods 5 (2013) 1010-1015, Title “Polypyrrole nanotube-polyaniline composite for DNA detection using methylene blue as intercalator”.
Response:
The following text was added to the Section 3.2 (line 216). The article suggested by reviewer was cited.
"Methylene blue intercalates DNA helix in the area reach with guanine residues. Besides, it can be electrostatically accumulated near minor grooves of DNA sequence by interaction with phosphate groups. Such interactions differ its electrochemical activity that can be monitored using conventional electrochemical tools. Mechanism and examples of application of Methylene blue in DNA sensors are considered in [43-45]."
Comment 6:
In the experimental section, the authors mention that the aniline concentration is 0.07 M. However, in the results and discussion, the authors say 7 mM of aniline. What is correct?
Response:
We thank to esteemed Referee for indication of this misprint. Aniline concentration is: 0.07 M. This has been corrected in revised manuscript.
Comment 7:
The author’s need to provide detailed information about the importance of polyaniline and their properties for different applications in the introduction part with support of the following few references (Sensors 2018, 18(11), 3926 and Indian Journal of Chemistry 50A (2011) 970-978).
We do not agree that polyaniline has found inadequate attention in Introduction. Among 36 references, 17 describe synthesis and application of polyaniline in sensors and biosensors. However, in accordance with the reviewer’ requirements, five more references were added. The total number of references exceeds that related to DNA-drug interactions which is the topic of investigation. Nevertheless, as the references suggested by reviewer are very interesting and valuable, we used them to substituted less important works. (refs. 31 and 25, respectively).
Comment 8:
In addition, there are several mistakes and typical errors, and kindly double-check the manuscript for example few: 2 Original: Methylene blue ; Revised: methylene in Line number 19
Response:
Spelling ‘Methylene blue” is now the same through the whole revised manuscript
Reviewer 2 Report
In this paper, authors proposed the novel DNA sensors that prepared layered coatings of biosensor with DNA entrapped between two independent PANI layers obtained by electropoly merization. It provided useful information. However, some problems need to be solved.
In abstract, the charge transfer resistance of the inner layer interface decreased with the doxorubicin concentration in the range from 1.0 pM to 0.1 μM (LOD 0.6 pM). The DNA sensor was tested for the analysis of spiked urine samples and showed satisfactory recovery in concentration range from 0.05–10 μM.
The question was samples. That was the real urine from human beings or artificial urine? If this DNA sensor was used in actual urine, what will be happen?
2.Line 218, “The conditions of doxorubicin determination were optimized to reach maximal sensitivity and lowest LOD (Table 1).”
Many equations were proposed to calculate the LOD value. What was the equation be used in this study? Why authors selected this equation?
3.Please describe the limitation, preparation and pretreatment of samples for the utilization of this DNA sensors by the information in Line 239 -246 and line 307-308
4.Ret, kΩ = (-0.26±0.06) +(0.32±0.01)×pC, M, R2 = 0.9942, n = 7
What was the mean of -0.26±0.06 and 0.32±0.01? Why only 7 samples were used? How about the replicates in this experiment?
5.In Table 2. Comparison of analytical characteristics of doxorubicin determination with electrochemical sensors and DNA sensors.
Please make a comment about the significant improvement about the LOD and concentration range? What was the reason that the significant improvement of this work?
6. 3.5. Real sample analysis, line 310
Applicability of the DNA sensor was confirmed by experiments with the samples of artificial urine spiked with doxorubicin. Only pH correction was required for sample testing.
If this DNA sensor was used for actual human urine, what will be happen? How about the performance? Is will be the same as artificial urine?

Author Response
We are grateful to the reviewer for most useful comments that allowed us to improve manuscript. All comments were taking into account in revised manuscript. The changes made are highlighted by red color. Response to the reviewer comments is below.
Comment 1:
The question was samples. That was the real urine from human beings or artificial urine? If this DNA sensor was used in actual urine, what will be happen?
Response:
The DNA sensor was tested on artificial urine, which content was provided in Experimental section. To avoid confusing, we added the word “artificial” to Abstract. Regarding real samples, we believe that the results will be similar as with artificial samples and we have obtained some preliminary estimates. However, such a conclusion can be made on much more number of samples and tests to take into account variability of the content of biological fluids. At the moment, we were intended to publish general principles of signal measurement and potential applicability of the biosensor for use in biological fluid.
Comment 2:
Many equations were proposed to calculate the LOD value. What was the equation be used in this study. Why authors selected this equation?
Response:
We have added the sentence “The LOD value was calculated for S/N = 3 criterion.” in revised manuscript just after the reference to Table 1 mentioned. Indeed, this is one of the most popular definitions of LOD, which is based on hypothesis of normal distribution of experimental data around real value and lack of bias of measurements. This assumption is one of the most frequently used in voltammetric and EIS measurements.
Comment 3:
Please describe the limitation, preparation and pre-treatment of samples for the utilization of these DNA sensors by the information in line 239-246 and line 307-308.
Response:
The following sentences were added after the lines 239-246:
"No specific requirements have been found for the use of Methylene blue solutions which can be prepared in advance and stored at room temperature for more than 2 weeks. To avoid mistakes, freshly prepared Methylene blue stock solution should be filtered to remove solid particles and impurities that could alter the dye concentration in contact with the biosensor."
The following sentences were added after the lines 307-308.
"When used after storage, the DNA sensor should be first incubated in Britton-Robinson buffer, pH 3.0, for 30-60 min. The incubation period can vary depending on the air humidity and storage temperature. The recovery of the biosensor is monitored by peak currents recorded prior to the contact with the drug solution and peak shape. "
Comment 4:
What was the mean of -0.26±0.06 and 0.32±0.01? Why only 7 samples were used? How about the replicates in this experiment?
Response:
Mean ± standard errors are presented for the slope and intercept of linear regression equation. All the measurements were performed in triplicate with individual sensors so that 21 samples were used. The appropriate amendments were introduced in the revised text (page 9, lines 296-299):
"The appropriate calibration curve is linear within the concentration range from 1.0 pM to 0.1 μM (Eq. 3). Mean ± standard error values were calculated for the slope and intercept of the curve. All the experimental points have been obtained in triplicate using individual sensors and analyte solutions."
Comment 5:
In Table 2. Comparison of analytical characteristics of doxorubicin determination with electrochemical sensors and DNA sensors. Please make a comment about the significant improvement about the LOD and concentration range. What was the reason that the significant improvement of this work?
Response:
The following sentences were added after the Table 2 mentioning (line 312):
"A significant improvement of the LOD value has been obtained due to dense contact of the DNA molecules with the PANI in the surface layer and its higher influence on intrinsic redox activity of the polymer matrix. Besides, contrary to the electropolymerization of aniline in the presence of DNA molecules, layer-by-layer immobilization allows choosing the conditions for DNA application and avoiding its contact with strong acids necessary for PANI electrochemical activity."
Comment 6:
3.5 Real sample analysis: If this sensor was used for actual human urine, what will be happen? How about performance? Is will be the same as artificial urine?
Response:
The following sentences were added to the manuscript (line 346):
Regarding real urine samples, it is believed the performance of the DNA sensor will be similar assuming higher deviation of the results caused from the personal variability of the urine content. Much more number of samples and measurements should be made to take into account possible complications and ensure patients in reliability of the results.
Round 2
Reviewer 2 Report
The content of revision has been improved